# AttackQA: Development and Adoption of a Dataset for Assisting Cybersecurity Operations using Fine-tuned and Open-Source LLMs

## Abstract

Retrieval-augmented generation (RAG) on specialized domain datasets has shown improved performance when large language models (LLMs) are fine-tuned for generating responses to user queries. In this study, we develop a cybersecurity question-answering (Q&A) dataset, called AttackQA, and employ it to build a RAG-based Q&A system designed for analysts in security operations centers. The dataset comprises 25,335 Q&A pairs, accompanied by rationales to facilitate fine-tuning and evaluation. 80% of the dataset was generated with help of a lightweight open-source LLM (LLama 3 8B), which produced over 1100 tokens per second with full 16-bit precision on specialized hardware. To ensure dataset quality, we fine-tuned LLama 3 70B to detect and reject low-quality Q&A pairs. In using the dataset for RAG, we demonstrate that fine-tuning open-source embeddings and LLMs can yield superior accuracy compared to OpenAI's state-of-the-art proprietary embedding and LLM (GPT-4o). Furthermore, we use Llama 3.1 405B as a judge to evaluate answer correctness, enabling the creation of a fully open-source, high-speed RAG and evaluation pipeline with an associated benchmark.

## 1 Introduction

Security operations centers (SOCs) house information security teams who are responsible for detecting, analyzing, and responding to cybersecurity incidents using a variety of tools, technologies, and processes. According to the Hershberger (2023) survey, the top challenges facing SOCs include a lack of expertise in security, too much time spent in investigating alerts, and a slow response time to advanced threats. To address those challenges and to enable quicker attack prevention and recovery, we propose an LLM-based question-answering (Q&A) system to help SOC analysts get quick answers to time-sensitive questions about cyberattacks. Our solution leverages entirely open-source large language models (LLMs) that are becoming increasingly powerful and, on domain-specific datasets, can be tuned to exceed the performance of proprietary LLMs that are many times as large.

We used the MITRE ATT&CK® (The MITRE Corporation, 2024) knowledge base of cyberattack techniques, tools, campaigns, detection approaches, and mitigation approaches to generate a Q&A dataset called AttackQA for use in Q&A systems or general-purpose chatbots. That knowledge base, grounded in real-world observations and updated biannually, was chosen because the ATT&CK® framework is widely adopted for cyber threat intelligence across the private sector, government, and the broader cybersecurity product and service community (Roy et al., 2023; AL-SADA et al., 2024; Al-Shaer et al., 2020). It is stored in an esoteric database format called Structured Threat Information Expression (STIX), making it ill-suited for direct use in Q&A systems, so we extracted the data and processed it in a way that makes it easier for training and inferencing with LLMs.

The structure of this paper and our approach are outlined in Fig. 1. The first phase involves the creation of the AttackQA dataset. Initially, we generated 28,686 Q&A pairs derived from the MITRE knowledge base. Subsequently, we fine-tuned Llama 3 70B to perform quality control (QC) on those Q&A pairs, retaining 25,335 high-quality examples. In the second phase, AttackQA was used to fine-tune both Microsoft's E5 Large V2 embedding (Wang et al., 2022) and Meta's Llama 3 8B LLM AI@Meta (2024) for retrieval-augmented generation (RAG). The accuracy of the results was

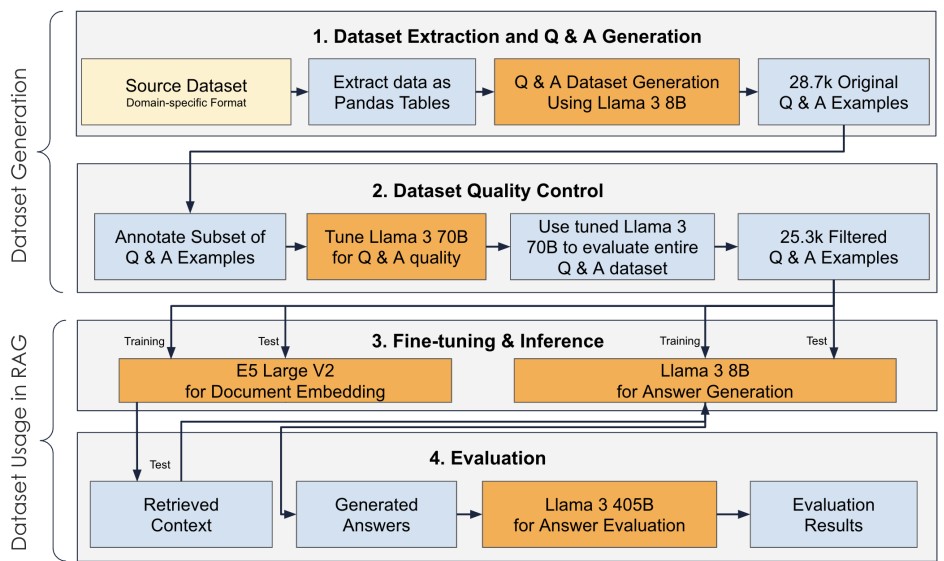

Figure 1: Illustration of dataset generation, quality control, and adoption in RAG

assessed using Llama 3 405B, leveraging a G-Eval (Confident AI, 2024) correctness metric within the DeepEval Framework.

In summary, our contributions are as follows:

- We demonstrate the use of a compact, open-source LLM (Llama 3 8B Instruct) to generate a high-quality question-answer dataset from the MITRE ATT&CK® knowledge base.

- We perform an evaluation that shows that a fine-tuned Llama 3 70B model is better than OpenAI's GPT-4o at identifying questions and answers that are of low quality, so they can be removed from AttackQA as part of an automated dataset quality control process.

- We demonstrate that fine-tuning an embedding model significantly enhances context recall in retrieval tasks, outperforming OpenAI's state-of-the-art (SOTA) embedding model, Text-Embedding-3-Large.

- We utilize Llama 3 405B as a judge to evaluate answer correctness. Using its evaluation scores, we found that fine-tuning Llama 3 8B as a generation model in RAG improves correctness, surpassing the performance of OpenAI's GPT-4o, which is many times as large.

- We developed an accurate and low-latency end-to-end RAG pipeline, utilizing fine-tuned open-source embeddings and LLMs to serve as a Q&A system to support security analysts.

By employing Llama 3 8B at speeds exceeding 1100 tokens/s (at full 16-bit precision), Llama 3 70B at over 550 tokens/s, and Llama 3 405B at 132 tokens/s, we were able to develop a highly responsive end-to-end solution for SOCs. Those model throughputs were achieved using the free SambaNova Cloud platform (SambaNova Systems, 2024) on specialized hardware (Prabhakar et al., 2024).

## 2 RELATED WORK

The use of LLMs for synthetic dataset generation, curation, and evaluation has been surveyed by Long et al. (2024). Although AttackQA is synthetically generated, it is grounded in the widely reputed MITRE ATT&CK® knowledge base. Levi et al. (2024) also fine-tune models on the MITRE ATT&CK® knowledge base, among other datasets, to teach models the relationships between source types. Similar to us, they too perform instruction tuning with chain of thought reasoning, but do not provide an approach to quality control their dataset or compare results with SOTA proprietary LLMs. The CyberQA dataset (priamai, 2024) is a related work in progress based off a dataset used in educational games. The use of LLMs in cybersecurity has been surveyed in (Motlagh et al., 2024).

The work of Hsieh et al. (2023) demonstrates that fine-tuning a 770M T5 model, using extracted rationales during the fine-tuning process, can outperform a few-shot prompted 540B PaLM model. Similarly, Zhang et al. (2024) fine-tuned the generation model within a RAG pipeline, enabling it to predict rationales alongside answers. Moreover, they fine-tuned the model using context that included both relevant and irrelevant (distractor) documents to improve its ability to answer questions. We employ the same set up in our work and show that greater accuracy improvements can be obtained by fine-tuning the embedding in addition to the LLM.

Yu et al. (2024) fine-tuned large language models to generate answers while simultaneously ranking context based on relevance. Izacard et al. (2022) introduced the pretrained LLM Atlas, specifically designed for retrieval-augmented generation, which achieved a 3% performance improvement over a 540B parameter model, despite having 50 times fewer parameters. Our fine-tuned embeddings perform well on ranking without requiring any unique training approaches and our generation models produce a 9% improvement on proprietary SOTA models that are much larger.

The fine-tuning of embeddings has been previously shown to enhance performance on tasks involving domain-specific datasets (Fabian et al., 2020). Synthetic dataset generation for the explicit purpose of fine-tuning such embeddings has been explored by Wang et al. (2024).

## 3 DATASET CREATION FOR Q&A

In this section, we describe our methodology for creating a Q&A dataset using the MITRE ATT&CK® knowledge base.

### 3.1 SUMMARY OF THE SOURCE DATA

The MITRE ATT&CK® knowledge base encompasses multiple categories of information, such as attack techniques, tactics, software tools used by attackers, well-known attacker groups, well-known attack campaigns executed by the attacker groups, mitigation strategies, and relationships.

The data for techniques, tactics, software, groups, campaigns, and mitigation approaches include a unique ID, name, description, and URL (an example is provided in Appendix A.1). From that data, we extracted the descriptions as text documents for use in Q&A tasks. The relationships table maps a *source type* to a *target type* via a *mapping type*. Source types include 'software', 'group', 'data component', 'mitigation strategy', and 'campaign', while target types consist of 'technique', 'software', and 'group'. The mapping types include 'uses', 'detects', 'mitigates', and 'attributed-to'. A mapping description was also provided and included in our set of Q&A documents.

Note that each instance of a source type is associated with a unique identifier. E.g., 'T1562' refers to an attack 'technique' and 'C0002' refers to an attack campaign. Specifying the identifier allows LLMs to make specific references with URLs to source data when answering questions.

AttackQA was generated using a mix of questions and answers generated by humans and LLMs. Each Q&A pair was derived from a single document, eliminating the need for multi-hop reasoning because comprehensive answers did not require information from multiple documents.

### 3.2 DOCUMENT PREPROCESSING

Newline characters were removed from within individual documents, ensuring that they appeared only between documents in the final retrieval context presented to the generation model. In all the documents, hyperlinks and special tags were replaced with plain text to ensure that neither the embeddings nor the generation models needed to process special tags that would not be encountered in questions and not be expected in answers.

From each document, one to three triplets of $\{question, thought, answer\}$ were generated, where *thought* represents the rationale necessary to accurately answer the question.

### 3.3 MANUAL Q&A GENERATION

Twenty percent of the Q&A pairs were generated by humans using heuristics embedded in code, relying solely on the relationships table for their creation.

Table 1: Examples of entries in AttackQA

| | Human-generated question, LLM-generated answer | LLM-generated question and answer |
|---|---|---|
| **question** | How does attack software 'S0467: TajMahal' use attack technique 'T1123: Audio Capture'? | How does Akira initially access victim environments? |
| **thought** | To answer the question, I need to understand how TajMahal, an attack software, utilizes the 'T1123: Audio Capture' technique. | To answer the question, I need to understand the initial access mechanism used by Akira as described in the document. |
| **answer** | TajMahal has the ability to capture VoiceIP application audio on an infected host. | Akira uses compromised credentials to access single-factor external access mechanisms such as VPNs for initial access |
| **document** | How attack software 'S0467: TajMahal' uses attack technique 'T1123: Audio Capture': TajMahal has the ability to capture VoiceIP application audio on an infected host. | Description of attack group 'G1024: Akira': Akira is a ransomware variant and ransomware deployment entity active since at least March 2023. Akira uses compromised credentials to access single-factor external access mechanisms such as VPNs for initial access, then various publicly-available tools and techniques for lateral movement. |
| **subject id** | T1123 | G1024 |
| **subject name** | Audio Capture | Akira |
| **subject type** | techniques | groups |
| **url** | https://attack.mitre.org/techniques/T1123 | https://attack.mitre.org/groups/G1024 |
| **source** | relationships uses software | groups |
| **references** | [{'source': 'T1123/TajMahal: https://attack.mitre.org/techniques/T1123', 'citation': 'TajMahal has the ability to capture VoiceIP application audio on an infected host.'}] | [{'source': 'groups/G1024/description: https://attack.mitre.org/groups/G1024', 'citation': 'Akira uses compromised credentials to access single-factor external access mechanisms such as VPNs for initial access'}] |
| **human question** | True | False |
| **human answer** | False | False |
| **field** | NaN | description |
| **relation id** | S0467 | NaN |
| **relation name** | TajMahal | NaN |

The human-generated questions resemble "What campaigns used attack technique 'T1562.001: Disable or Modify Tools'?" The corresponding answers resemble "The campaigns that used attack technique 'T1562.001: Disable or Modify Tools' were: 'C0002: Night Dragon', 'C0024: SolarWinds Compromise', 'C0028: 2015 Ukraine Electric Power Attack', 'C0029: Cutting Edge'". Because that answer was not available in any single document in the source dataset, we synthetically generated a document to match the answer. That ensured that the full list of relationships for a given source type, target type, and mapping type were available in a single document for ease of retrieval. To generate the document, it was sufficient to query the relationships table, filtering on the relevant entities (e.g., campaigns, software, techniques, etc.). The questions were generated to ensure comprehensive coverage of source types, target types, and mapping types. Notably, no list of relationships was long enough to cause the answer to exceed 1000 tokens in length.

## 3.4 LLM-BASED AUTOMATED Q&A GENERATION

Utilizing Llama 3 8B Instruct AI@Meta (2024), we generated Q&A based on the processed documents. To speed up the process, we leveraged SambaNova's free cloud offering (SambaNova Sys-

tems, 2024), which runs Llama 3 8B at over 1100 tokens/s with full 16 bit precision Kerner (2024). Although we experimented with Llama 3 70B, the results were not significantly different, leading us to continue with Llama 3 8B. In some instances, Llama 3 70B produced overly verbose responses that were difficult to quickly comprehend. Examples of dataset entries generated using Llama 3 8B are provided in Table 1.

Half of the Q&A pairs comprised human-generated questions and LLM-generated answers. The questions encompassed those that we anticipated end users would most likely ask and were structured as follows: "Describe X", "How can X detect Y?", "How can X mitigate Y?", and "How does attack software X use attack technique Y?", where X and Y were subject names, similar to those listed in Table 1. Each of those questions could be answered by a specific document in AttackQA. Therefore, the generation model simply had to summarize that document in response to the question.

The remaining 30% of Q&A pairs were generated using Llama 3 8B Instruct, where the question, answer, and rationale were all derived from a given document. Depending on the document length, up to three sets of $\{question, thought, answer, references\}$ were generated in a single LLM completion. The precise prompt utilized for that generation is given in Appendix A.4.1.

## 3.5 ENSURING QUALITY OF LLM-GENERATED DATA

Through careful prompt engineering and post-processing of special characters (e.g., the '\' character in file paths), we achieved a 99% success rate in ensuring that Llama 3 8B produced valid JSON in the required format. That was despite the present lack of a JSON mode in the SambaNova API.

To ensure the quality of the LLM-generated questions and answers, we employed three strategies:

1. We mandated that the LLM generate a citation for each question-answer-rationale entry that included verbatim text from the document supporting the answer. That requirement ensured that the entry was grounded in the source material.

2. All instances of duplicated questions were removed from the dataset, allowing the remaining questions to act as a unique index. The reason is that the presence of duplicate questions implies that the same inquiry could pertain to two or more different documents, indicating that the question lacked specificity to any particular document.

3. We leveraged an independent grader LLM to grade each entry in the dataset on the quality of the question and answer. We refer to that LLM as the quality control (QC) LLM and the remainder of this subsection will describe that automated curation approach.

### 3.5.1 CRITERIA FOR ASSESSING Q&A QUALITY

We employed the G-Eval metric Confident AI, 2024 in the DeepEval framework to facilitate automatic curation using the quality control (QC) LLM. The G-Eval metric is a custom metric that scores each Q&A pair by assessing it in conjunction with the retrieved context. In addition to generating a score, the prompts formulated by DeepEval require the LLM to provide a rationale for the assigned score. The scoring is based on specific evaluation steps that establish the scoring criteria. We developed distinct metrics for assessing both question quality and answer quality.

For the assessment of question quality, we evaluated whether the questions were ambiguous, failed to reference specific topics in the context, referred to topics not present in the context, or were so broad that they could be answered with information outside of the provided context.

For the assessment of answer quality, we examined whether the answers were irrelevant to the question, did not reference pertinent content from the context, lacked comprehensiveness, were vague, or included information not contained within the context.

In DeepEval, the model is prompted to generate a score ranging from 0 to 10. However, that score is subsequently divided by 10 during post-processing to yield a normalized range from 0 to 1.

### 3.5.2 FINE-TUNING THE QC LLM

We manually annotated 400 Q&A pairs, assigning scores along with reasons to justify the scores. We kept 80 of those pairs as a hold-out validation set to evaluate models ensuring that the score

distributions were preserved (see Appendix A.2). A high score indicates that the Q&A pair is of high quality and worth retaining, whereas pairs with low scores were filtered out.

According to our annotations, a score of 1 denoted a perfect example, while scores of 0.8 and 0.9 were deemed acceptable, but not flawless. Any score below 0.7 indicated significant quality issues and the example was marked for removal from the dataset. We classified examples from the dataset that are deemed worth retaining (i.e., $score > 0.7$) as positives, and those that are better suited for removal (i.e., $score \leq 0.7$) as negatives. Under this classification, 31% of the training set and 32.5% of the validation set were identified as negatives.

At the time of writing, OpenAI's GPT-4o is recognized as a SOTA proprietary model. However, its ability to judge data quality was found to be inadequate. Despite the validation set containing 26 negatives, GPT-4o identified only 2 as negatives, while Llama 3 70B did not predict any negatives. This resulted in $(precision, recall)$ values of $(79\%, 98\%)$ for GPT-4o and $(71\%, 100\%)$ for Llama 3 70B. Neither model demonstrated proficiency in predicting negatives, and the high recall values were largely attributable to the trivial strategy of predicting all examples as positives.

To ensure that the LLM-based QC process would yield a filtered dataset with a high precision (in which case most predicted positives turn out to be true positives), we opted to fine tune Llama 3 70B on the annotated training set. Full parameter tuning was conducted using the AdamW (Loshchilov & Hutter, 2019) optimizer with a fixed learning rate of $10^{(-5)}$ and a weight decay of $0.1$. As previously mentioned, the base model achieved a perfect recall due to its failure to predict any negatives. However, as we fine-tuned the model over additional steps, we observed a decrease in recall accompanied by an increase in precision (see Appendix A.3 for more details). The final QC assessment was performed using a checkpoint of Llama 3 70B that was fine-tuned for 65 steps, resulting in a validation precision of 84.2% and a recall of 89%. By prioritizing precision over recall, we chose a model that was better at detecting negatives (bad examples) and minimizing their presence in the final dataset. The consequent reduction in recall led to the exclusion of some positive examples (along with the negatives) from the final dataset, a trade-off that ensured quality over quantity.

### 3.6 Dataset Summary

After performing QC, AttackQA contained the 25,335 high-quality Q&A pairs covering 17,760 unique documents. As measured by the cl100k_base tokenizer with Tiktoken (Open AI, 2024), the largest document had a length of 3,103 tokens. The smallest document had a length of 15 tokens and the average document length was 75 tokens.

In using the dataset, some of the documents may need to be chunked for use with models with small context windows. Note that only 104 out of 17,760 (0.6%) of documents have greater than 500 tokens in length and the rest could be used directly with a model of 4096 context length. In the analyses presented in the following sections, we did not chunk any of the documents the open-source LLMs we used had context windows of length 8192 tokens.

## 4 Model Fine-tuning for RAG

A basic RAG framework is illustrated in Fig. 2. In this section, we used AttackQA to fine-tune the LLMs and embeddings to improve answer accuracy in that framework.

Prior to any user interaction, documents are embedded by the embedding model and stored in a vector database. When a user asks a question, the question is embedded by the same embedding model and $k$ documents relevant to the question are retrieved from the vector database. In our analysis, we retrieve the $k$ documents based on the similarity between their embeddings and the question's embeddings. The $k$ documents are then presented to the generation model in a prompt that asks the model to answer the user's questions using information from the documents. The answer is then returned to the user.

Note that there a more complex implementations of the framework involving multiple generation models, re-ranking models, and multiple types of data stores. Such implementations are beyond on the scope of this analysis, which seeks to measure the contributions of fine-tuning individual models on the overall accuracy.

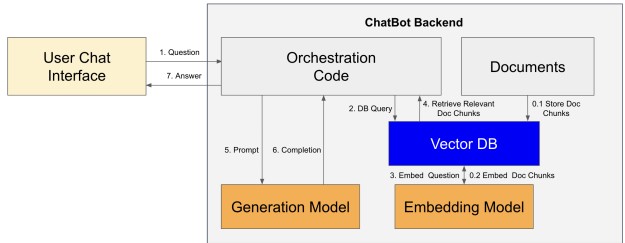

Figure 2: A basic retrieval augmented generation (RAG) framework

## 4.1 TRAINING AND EVALUATION SPLIT

We split the $25,335$ Q&A pairs into a training set ($90\%$) and an evaluation set ($10\%$) using uniform random sampling. Similar to Zhang et al. (2024), we ensured that all documents were represented in the training set so that the trained models would be familiar with the knowledge base from which questions would be asked. However, the questions in the evaluation set were not present in the training set. That resembles a live production usage setting, in which the end user wants to ask questions of a dataset, and the chatbot is familiar with the source documents but may not have previously seen the questions.

When fine-tuning the models, we used 10% of the training set for validation to ensure that we would eventually evaluate on a checkpoint that was not over-fitting on the training set.

## 4.2 EMBEDDING MODEL

We performed full parameter fine-tuning for two epochs using Microsoft's E5 Large V2 embedding model (Wang et al., 2022) on one Reconfigurable Dataflow Unit (RDU) (Prabhakar et al., 2024). The model has 335M parameters and encodes up to 512 tokens into an embedding of length 1024. The training dataset comprised of a list of questions from the training set. For each question, a list of positive documents (containing the answer) and negative documents (not containing the answer) were provided. By construction of the dataset, only one positive document existed for each question (since the question was generated from that document).

Having negative documents helps the model learn to distinguish between relevant and irrelevant documents for a given question using contrastive learning (Chopra et al., 2005). The negative documents were randomly sampled from a set that excluded documents whose entities were related to the entity associated with the question. That ensured that the answers could not accidentally be obtained from the negative documents, leading to poor contrastive learning. Related entities can be identified in the MITRE dataset based on their IDs (e.g., T1562.001 and T1562.002 are related techniques and should not be included in negative documents for any question relating to T1562.xxx).

## 4.3 GENERATION MODEL

Llama 3 8B Instruct was fine-tuned using full-parameter fine-tuning for two epochs (batch size of 64) on 8 RDUs. the same questions that were used to train the embedding model, but the dataset preparation for training was different. For each question, a set of $k$ documents were retrieved using Microsoft's E5 Large V2 embedding. We denote the retrieved set of documents by $\boldsymbol{d}(k) = \{d_1, ..., d_k\}$.

Let $d^*$ denote the golden document, which contained the answer to the question. For all the 23,263 Q&A pairs in the training set, we used post-processing to ensure that their corresponding $d^* \in \boldsymbol{d}$. Specifically, when $\boldsymbol{d}$ did not contain $d^*$ from the retrieval, we used code to replace $d_k$ with $d^*$. The remaining $k-1$ documents, $d_i \neq d^*$, were distractor documents that the LLM would need to learn to ignore because the model would be presented with all $k$ documents even in production. We shuffled the $k$ documents in $\boldsymbol{d}$ to ensure that the model did not learn to pick $d^*$ based on its retrieval rank in $\boldsymbol{d}$, and focused on the contents of the documents instead of the ordering.

Table 2: Context recall in top $k$ documents for retrieval

| Top $k$ | Base E5 Large V2 | Tuned E5 Large V2 | OpenAI TE-3-Large |
|---|---|---|---|
| $k = 10$ | 72.8% | 93.29% | 83.58% |
| $k = 5$ | 69.84% | 92.18% | 80.85% |
| $k = 1$ | 57.6% | 81.48% | 65.1% |

Each prompt comprised of an instruction with a one-shot example, and the retrieved list of documents, $\boldsymbol{d}(k)$, with $k = 5$. The completions included a thought, answer, and references. The thought was included to ensure that the model's answers were well-reasoned and the references ensured that the right document in the $\boldsymbol{d}$ was being used in answering the questions. An example of a prompt-completion pair is given in Appendix A.4.2.

We augmented the training set with 3,323 additional examples (amounting to one-eighth of the total training set) to train the model not to hallucinate. In doing so, we re-used questions in the training set for which $d^* \notin \boldsymbol{d}$ and had the completions modified with answers that read "I am sorry, I do not have the answer to the question." and an empty references list.

## 5 MODEL EVALUATION

In this section we present the approach to and results of our model evaluations. All results are presented on the hold-out evaluation set comprising 2,533 examples.

### 5.1 RETRIEVAL MODEL

The retrieval component of the pipeline refers to steps 2-4 in Fig 2. We evaluated that component using the context recall metric, which captures whether or not the retrieved context contains the golden document. Based on a metric, which in our analysis was the similarity metric, a vector database can be configured to return the top $k$ results, $\boldsymbol{d}(k, q_i)$, for a given query $q_i$. We seek a metric to check if $d^*(q_i) \in \boldsymbol{d}(k, q_i)$ for all $q_i$ in the evaluation set. For an evaluation set of $N$ queries, the context recall (denoted by $R$) is computed as follows:

$$R(k) = \frac{1}{N} \sum_{i=1}^{N} \mathbf{1}_{d^*(q_i) \in \boldsymbol{d}(k, q_i)} \tag{1}$$

The results for $k \in \{1, 5, 10\}$ are summarized in Table 2. In all cases, the fine-tuned E5 Large V2 model significantly outperformed both the base E5 Large V2 model and Open AI's SOTA embedding model, Text Embedding 3 Large. The reason is that the dataset contained a lot of domain-specific jargon relating to cybersecurity that the base embedding models were not able to encode. Furthermore, the tuned embedding returned $d^*$ in top 5 ranks in $\boldsymbol{d}$ 92.18% of the time, indicating that a re-ranker model would not have been necessary to bump $d^*$ from the top 10 to the top 5. Finally, the tuned embedding produced $d^*$ at the top rank 81.48% of the time, indicating strong ranking ability.

### 5.2 GENERATION MODEL

Because the answers generated by the generation models are all free-form text, it was difficult to come up with an objective evaluation of their correctness. Objective metrics like Bleu (Papineni et al., 2002) and Rouge (Lin, 2004) perform N-gram comparisons between expected and actual answers and may not recognize when the two are semantically equivalent if they use different words. For that reason, we used an LLM-as-a-judge to score the answers for correctness.

Once again, we used the G-Eval metric with DeepEval to score answers and provide reasons for the scores. With regard to evaluation criteria, we required that the generated answers be penalized for correctness if they 1) contradicted the true answer, 2) omitted details from the true answer that were relevant to the question, and 3) included irrelevant detail that were not present in the true answer.

Table 3: Pipeline evaluation of different embedding and generation model configurations

| | TE-3-L GPT-4o | Base Emb Base Gen | Base Emb Tuned Gen | Base Emb GPT-4o | Tuned Emb Base Gen | Tuned Emb Tuned Gen | Tuned Emb GPT-4o |
|---|---|---|---|---|---|---|---|
| **% Context Recall at k=5** | 80.85 | 69.84 | 69.84 | 69.84 | **92.18** | **92.18** | **92.18** |
| **% Answer parsing success** | 99.96 | 98.18 | 99.88 | 99.96 | 99.53 | **100.00** | 99.84 |
| **% Correct reference** | 85.47 | 74.97 | 74.97 | 75.96 | 91.04 | 88.91 | **92.10** |
| **% Mean Correctness (soft)** | 81.58 | 75.10 | 78.49 | 82.27 | 78.60 | **88.12** | 82.87 |
| **% Mean Correctness (hard)** | 72.38 | 60.97 | 70.24 | 63.99 | 76.03 | **86.07** | 79.08 |

We used Llama 3.1 405B (Zhou et al., 2024) for the aforementioned evaluation with DeepEval for its SOTA judging ability (Raju, 2024), speed, and cost (it is provided at 132 tokens/s for free by SambaNova Cloud). Seven combinations of embedding and generation models in the RAG frame-work were evaluated and the evaluation results are summarized in Table 3. "Base Emb" and "Tuned Emb" refer to the base and fine-tuned versions of E5 Large V2 embedding model, respectively. "Base Gen" and "Tuned Gen" refer to the base and fine-tuned versions of Llama 3 8B generation model, respectively. TE-3-L refers to Open AI's SOTA 'Text Embedding 3 Large' model.

The first row of Table 3 recaps the context recall from Table 2 for $k = 5$ to show how the other results may be impacted by it. The answer parsing success relates to the generation model's ability to produce JSON-formatted answers in the required format. That all combinations have at least a 98% parsing success indicates that the prompts were adequately engineered. "% Correct reference" refers to the number of examples for which the correct reference was produced by the generation model. The references comprise URLs that are included in the retrieved context.

Two correctness scores are provided and both use the same G-Eval metric with Llama 3 405B. In the case of "mean correctness (soft)", if $d^*(q_i) \notin \boldsymbol{d}(k, q_i)$ for any $q_i$ and the generated answer is "I am sorry, I do not have the answer to the question," then we mark the answer as 100% accurate. That metric compensates for an inaccurate embedding in the retrieval component, explaining why it makes no difference to the result when we switch the embedding from base to tuned while keep the same base generation model (either Base Gen or GPT-4o).

The "mean correctness (hard)" metric requires that the generated answer match the true answer, regardless of the embedding's retrieval accuracy. No concessions are given for the generation model not admitting to knowing the answer. Therefore, soft correctness scores are higher because the answers that were incorrect by hard correctness were forgiven my soft correctness.

The biggest gain on hard correctness, an improvement of 26 percentage points, was achieved when going from a Base Emb/Base Gen combination to a Tuned Emb/Tuned Gen combination. An im-provement of 16 percentage points was achieved by swapping out the base embedding with a tuned one, for the same generation model.

Tuning the generation model allows it to correctly answer questions even if the answer is not present in the retrieved context leading to an improvement of 10 percentage points when going from a base generation model to a tuned generation model while keeping the embedding the same.

The first column in Table 3 refers to a solution using Open AI's SOTA embedding and generation models. On hard correctness, that combination outperforms all other combinations that use the base embedding, but it underperforms those that use the tuned embedding. Therefore, tuning the embedding model is essential to beating proprietary SOTA models using open source SOTA models on our evaluation set.

## 5.3 CASE STUDY

In this section, we present an evaluation case study to take a deeper look at the results. Additional case studies are included in Appendix A.5. The case study, presented in Table 4, refers to a specific Q&A pair in the evaluation set. That pair was among the 30% of all pairs for which the question and the answer were generated from the document using LLama 3 8B.

The generated answers came from either GPT 4-o, Llama 3 8B (base), or Llama 3 8B (fine-tuned), as specified in the column headers. The row $d^*$ rank in context refers to the ranked position in $\boldsymbol{d}$ when

Table 4: Evaluation Case Study: What is the purpose of KOPILUWAK?

| | | | | | |
|---|---|---|---|---|---|
| **Question** | What is the purpose of KOPILUWAK? | | | | |
| **True answer** | KOPILUWAK has been used for victim profiling and C2 since at least 2017. | | | | |
| **Document** | Description of attack software 'S1075: KOPILUWAK': KOPILUWAK is a JavaScript-based reconnaissance tool that has been used for victim profiling and C2 since at least 2017. | | | | |
| | **Base Emb, Base Gen** | **Base Emb, Tuned Gen** | **Tuned Emb, Base Gen** | **Tuned Emb, Tuned Gen** | **OpenAI Emb, GPT 4-o** |
| **Generated answer** | I am sorry, I do not have the answer to the question. | KOPILUWAK is a command-line utility used to compress and encrypt data collected prior to exfiltration, such as tar. | KOPILUWAK is a JavaScript-based reconnaissance tool that has been used for victim profiling and C2 since at least 2017. | KOPILUWAK has been used for victim profiling and C2 since at least 2017. | I am sorry, I do not have the answer to the question. |
| $d^*$ **rank in context** | $d^* \notin \boldsymbol{d}$ | $d^* \notin \boldsymbol{d}$ | 1 | 1 | $d^* \notin \boldsymbol{d}$ |
| **Correctness score (hard)** | 0.0 | 0.0 | 0.8 | 1.0 | 0.0 |
| **Correctness reason** | The actual output does not provide any relevant information about KOPILUWAK, whereas the expected output provides specific details about its purpose. | The actual output provides unrelated information about data compression and encryption, whereas the expected output mentions its use for victim profiling and C2 since 2017, indicating a contradiction. | The actual output provides extra information about the programming language of KOPILUWAK, which is not present in the expected output. However, it does convey the same core information as the expected output regarding the purpose of KOPILUWAK. | The actual output matches the expected output exactly, with no omissions or irrelevant details. | The actual output does not provide any relevant information about KOPILUWAK, whereas the expected output provides specific details about its purpose. |

the retrieval succeeds (otherwise it reads $d^* \notin \boldsymbol{d}$). The configurations with the tuned embedding both returned $d^*$ at the highest rank, which is desirable. The others did not return $d^*$ at all, likely because they were not able to properly embed the domain-specific term "KOPILUWAK". Consequently, the configurations with the tuned embeddings produced correct answers, whereas the others did not.

The hard correctness score and reason were both produced by Llama 3 405B, which we used for judging. It produced a score of 0.8 for the "Tuned Emb, Base Gen" configuration and its reasoning is clear that the answer includes irrelevant details. The scores for the "Tuned Emb, Base Gen" and Open AI configurations would have been set to 1 for soft correctness for their admission to not knowing the answer. The "Base Emb, Tuned Gen" configuration, however, would have received a soft correctness score of 0 for hallucinating.

## 6 CONCLUSION

In this work, we created a Q&A dataset based on the MITRE ATT&CK® database of cyberattack techniques, software, campaigns, mitigation approaches, and detection approaches. The dataset, AttackQA, can be used to train models and create a chatbot to help security operations center analysts decrease their time to mitigate cyberattacks by giving them easy and fast answers to questions that they may have about the attacks. We presented an approach to automatically generate data and perform quality control on that data using SOTA open-source LLMs.

We evaluated a RAG pipeline using our dataset and showed that fine-tuning both the generation and embedding models can lead to an increase in hard accuracy of 26 percentage points. Fine-tuning the embedding model alone can lead to an improvement of 16 percentage points. Finally, fine-tuning the generation model alone, as proposed by Zhang et al. (2024), leads to an accuracy improvement of 10 percentage points. Open AI's SOTA models produced high accuracy but could be outperformed by tuning openly available embedding models. Even when GPT 4-o was combined with our tuned embeddings, it underperformed a fine-tuned Llama 3 8B model, which was many times smaller. That is particularly advantageous when deploying Q & A systems to hardware-constrained resources; LLama 3 8B can run on a commodity laptop, which can be air-gapped for security purposes. GPT 4-o, however would require an API call because it is too large to run on commodity hardware.

The AttackQA dataset and associated benchmarking code are made openly available (author, 2024). Researchers may use the dataset to evaluate several other open-source embeddings and LLMs (e.g., Qwen, Mistral, etc.) that may outperform Llama 3 8B. However, since AttackQA contains domain-specific jargon, we expect that the best performing models would need to be tuned on the dataset.

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

# A APPENDIX

## A.1 SUMMARY AND EXAMPLES FROM THE MITRE SOURCE DATASET

Table 5: Types of Entries in the MITRE ATT&CK® knowledge base

| Dataset | Description | # of Entries |
|---|---|---|
| Techniques | Attack techniques | 637 |
| Tactics | Attack tactics. One attack technique may employ multiple tactics | 14 |
| Software | Software tools used by cyber attackers when executing techniques | 677 |
| Groups | Names of well-known attacker groups responsible for well-known attack campaigns | 148 |
| Campaigns | Attack campaigns that may leverage one or more techniques | 28 |
| Mitigations | Mitigation strategies used for the attacks | 43 |
| Relationships | Relationships between techniques, software, groups, campaigns, detection and mitigation approaches | 17324 |

Table 5 provides a summary of the classes of attack information contained in the MITRE knowledge base. Two examples of entries extracted from the MTIRE knowledge base are also presented in this section. The first is an example of a software tool used by attackers and is presented in Table 6. There are 677 such entries in the MITRE knowledge base. Entries for techniques, tactics, groups, campaigns, and mitigation approaches also include unique ID, name, description, and URL.

Table 6: Example software table entry from source data

| Field | Value |
|---|---|
| ID | S0066 |
| name | 3PARA RAT |
| description | [3PARA RAT](https://attack.mitre.org/software/S0066 ) is a remote access tool (RAT) programmed in C++ that has been used by [Putter Panda](https://attack.mitre.org/groups/G0024 ). (Citation: CrowdStrike Putter Panda) |
| url | https://attack.mitre.org/software/S0066 |
| contributors | NaN |
| platforms | Windows |
| aliases | |
| type | malware |
| relationship citations | (Citation: CrowdStrike Putter Panda), (Citation: CrowdStrike Putter Panda) |

In creating AttackQA, we preprocess descriptions like "[3PARA RAT](https://attack.mitre.org/software/S0066 ) is a remote access tool (RAT) programmed in C++ that has been used by [Putter Panda](https://attack.mitre.org/groups/G0024 ). (Citation: CrowdStrike Putter Panda)" to "3PARA RAT is a remote access tool (RAT) programmed in C++ that has been used by Putter Panda."

An example of a relationship entry in the MITRE knowledge base is presented in Table 7.

Table 7: Example relationships table entry from source data

| Field | Value |
|---|---|
| source ID | M1036 |
| source name | Account Use Policies |
| source type | mitigation |
| mapping type | mitigates |
| target ID | T1110 |
| target name | Brute Force |
| target type | technique |
| mapping description | Set account lockout policies after a certain number of failed login attempts to prevent passwords from being guessed. Too strict a policy may create a denial of service condition and render environments un-usable, with all accounts used in the brute force being locked-out. Use conditional access policies to block logins from non-compliant devices or from outside defined organization IP ranges.(Citation: Microsoft Common Conditional Access Policies) |

## A.2 DISTRIBUTION OF SCORES IN QUALITY CONTROL DATASET

The distribution of scores in the dataset used for training an LLM to perform quality control on the LLM generated Q & A examples is illustrated in Fig. 3 Quality Control dataset.

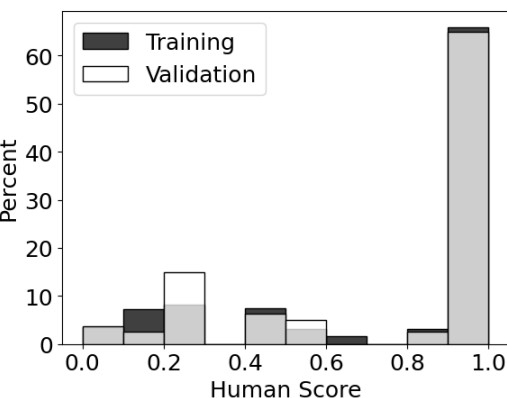

Figure 3: Distribution of scores in the manually-annotated QC dataset

## A.3 PRECISION-RECALL CURVE FOR QUALITY CONTROL LLM

The precision and recall for multiple fine-tuning steps are illustrated in Fig. 4. It can be seen that the highest precision was achieved at 65 steps, beyond which tuning led to a significant drop in precision without much improvement in recall.

## A.4 EXAMPLE PROMPTS

In this section, we present the exact prompts that were used for dataset generation and for the RAG application.

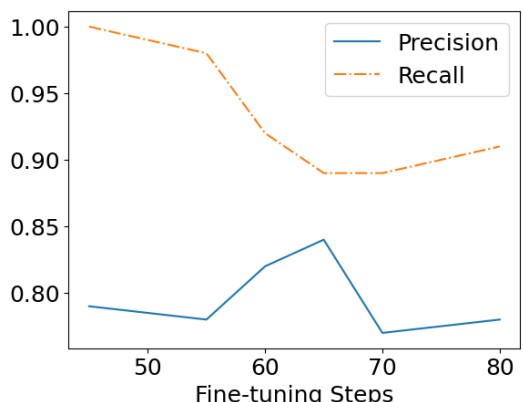

Figure 4: Precision & recall of QC LLM on annotated validation set

### A.4.1 PROMPTS USED FOR DATASET GENERATION

The Python function used to construct the prompts for dataset generation using LLMs is presented below. Valid JSON, containing up to three entries, was generated 99% of the time even when the API did not support token sampling constraints for JSON outputs. The prompt template is customized for Llama 3 models.

```python
def get_prompt_for_doc(doc, count="three sets"):
    prompt =
    """<|begin_of_text|><|start_header_id|>system<|end_header_id|>
    You are a JSON generator who generates machine-readable
    JSON<|eot_id|><|start_header_id|>user<|end_header_id|>
            Based on the following document, follow the
            instruction below
            Document:
            %s
            Instruction:
            Generate %s of unique question, thought, answer, and
            references from the above document in the following
            JSON format. The answers must avoid words that are not
            specific (e.g., "many", "several", "few", etc.). The
            answers must contain specific, verbose,
            self-contained, grammatically correct sentences that
            answer the question comprehensively. The answers must
            strictly contain content from the document and no
            content from outside the document. There may be
            multiple references that contain verbatim text from
            the document to support the answers.
            JSON format:
            [
                {
                    "question": "<generated question>",
                    "thought": "<generated thought on what is
                    needed to answer the question. Start with 'To
                    answer the question, I need'>",
                    "answer": "<generated answer>",
                    "references": [
                        "<verbatim text from document that
                        supports the answer>",
                        "<verbatim text from document that
                        supports the answer>"
```

```
                    ]
                }
            ]
            The first character of the response must be '[' and
            the last character must be ']'. No header text should
            be included.
            <|eot_id|><|start_header_id|>JSON
            list<|end_header_id|>
            """%(doc, count)
    return prompt
```

### A.4.2 PROMPTS FOR GENERATING ANSWERS IN RAG

The RAG prompt contains instructions, a one-shot example to illustrate the required response format, and the entire list of $k = 5$ documents from the retrieval model. The prompt template was used for both fine-tuning the model and for inference on the base or fine-tuned models. The tags in the example prompt, which are specific to Llama 3, were removed when performing inference using GPT-4o.

**Prompt:**

---

<—begin_of_text—><—start_header_id—>system<—end_header_id—>
You are an assistant for generating JSON formatted responses
<—eot_id—><—start_header_id—>user<—end_header_id—>
Respond with a JSON dictionary that includes a thought, answer, and references
The answer must contain text obtained strictly from the given documents.
Avoid any text that is not in the given documents.
Answer using concise, self-contained, grammatically complete sentences.
The answer must be a string with less than four sentences.
Do not mention the documents by number or the context in the answers.
Answer the question strictly using the provided documents.
If you cannot answer the question using the documents, the answer should be "I am sorry, I do not have the answer to the question."
Along with the answer, include a thought that begins with "To answer the question, I need".
The references must contain URLs that exactly match the full URLs in the document headers relevant to by the answer.
There may be multiple references in the references list.
Follow the example below:

Document 1: https://attack.mitre.org/techniques/T1562/001

The campaigns that used attack technique 'T1562.001: Disable or Modify Tools' were: 'C0002: Night Dragon', 'C0024: SolarWinds Compromise', 'C0028: 2015 Ukraine Electric Power Attack', 'C0029: Cutting Edge'"

Document 2: https://attack.mitre.org/techniques/T1562/002

The campaigns that used attack technique 'T1562.002: Disable Windows Event Logging' were: 'C0024: SolarWinds Compromise', 'C0025: 2016 Ukraine Electric Power Attack'

Document 3: https://attack.mitre.org/techniques/T1070/001

The campaigns that used attack technique 'T1070.001: Clear Windows Event Logs' were: 'C0014: Operation Wocao'

Question: What campaigns used attack technique 'T1562.002: Disable Windows Event Logging'?
Response:
{

"thought": "To answer the question, I need to know what campaigns used attack technique 'T1562.002: Disable Windows Event Logging'. The answer is contained in the provided document with URL 'https://attack.mitre.org/techniques/T1562/002'.",
"answer": "The campaigns that used attack technique 'T1562.002: Disable Windows Event Logging' were: 'C0024: SolarWinds Compromise', 'C0025: 2016 Ukraine Electric Power Attack'",
"references": {["https://attack.mitre.org/techniques/T1562/002"]}
}

Document 1: https://attack.mitre.org/techniques/T1539
How data component 'Process Access' can be used to detect attack technique 'T1539: Steal Web Session Cookie':
Monitor for attempts by programs to inject into or dump browser process memory.

Document 2:
https://attack.mitre.org/techniques/T1539
The following 2 data components can be used to detect attack technique 'T1539: Steal Web Session Cookie': File Access, Process Access

Document 3: https://attack.mitre.org/techniques/T1539
The software procedures that use attack technique 'T1539: Steal Web Session Cookie' are: 'S0467: TajMahal', 'S0492: CookieMiner', 'S0531: Grandoreiro', 'S0568: EVILNUM', 'S0631: Chaes', 'S0650: QakBot', 'S0657: BLUELIGHT', 'S0658: XCSSET'

Document 4: https://attack.mitre.org/techniques/T1539
Tactics used in attack technique 'T1539: Steal Web Session Cookie': Credential Access

Document 5: https://attack.mitre.org/techniques/T1539
Description of attack technique 'T1539: Steal Web Session Cookie': An adversary may steal web application or service session cookies and use them to gain access to web applications or Internet services as an authenticated user without needing credentials. Web applications and services often use session cookies as an authentication token after a user has authenticated to a website. Cookies are often valid for an extended period of time, even if the web application is not actively used. Cookies can be found on disk, in the process memory of the browser, and in network traffic to remote systems. Additionally, other applications on the targets machine might store sensitive authentication cookies in memory (e.g. apps which authenticate to cloud services). Session cookies can be used to bypasses some multi-factor authentication protocols. There are several examples of malware targeting cookies from web browsers on the local system. Adversaries may also steal cookies by injecting malicious JavaScript content into websites or relying on User Execution by tricking victims into running malicious JavaScript in their browser. There are also open source frameworks such as 'Evilginx2' and 'Muraena' that can gather session cookies through a malicious proxy (e.g., Adversary-in-the-Middle) that can be set up by an adversary and used in phishing campaigns. After an adversary acquires a valid cookie, they can then perform a Web Session Cookie technique to login to the corresponding web application.

Question: What is a potential indicator of the 'T1539: Steal Web Session Cookie' attack technique?
The response must be formatted as a JSON instance that conforms to the JSON schema above.
No text should appear before or after the JSON instance.
Response:
<—eot_id—><—start_header_id—>machine-readable JSON<—end_header_id—>

**Completion:**

{
"thought": "To answer the question, I need to understand how to detect the 'T1539: Steal Web Session Cookie' attack technique. The answer is contained in the provided document with URL 'https://attack.mitre.org/techniques/T1539'.",
"answer": "A potential indicator of the 'T1539: Steal Web Session Cookie' attack technique is monitoring for attempts by programs to inject into or dump browser process memory.",

"references": ["https://attack.mitre.org/techniques/T1539"]
}

The above completion contains a $thought$, $answer$, and $references$. It is only used for fine-tuning the generation model and is constructed using fields from AttackQA. The $thought$ describes the rationale and is included to help the model learn to find the right document and use it to answer the question.

## A.5 ADDITIONAL CASE STUDIES

Table 8 presents a case study in which all the embeddings find the $d^*$. Although the OpenAI embedding achieved a higher rank of 1 (all others had a rank of 3), GPT-4o generated a less accurate answer than the tuned generation Llama 3 8B. Like GPT-4o, even the base Llama 3 8B failed to mention that "testing and debugging" are purposes of the 'Office Test' registry key. That is despite the fact that the name of the key implies the purpose and the purpose is explicitly stated in $d^*$. That case study highlights that tuning the generation model with rationales can help improve reasoning.

Table 8: Evaluation Case Study: What is the purpose of the 'Office Test' Registry key?

| Question | What is the purpose of the 'Office Test' Registry key? | | | | |
|---|---|---|---|---|---|
| True answer | The 'Office Test' Registry key is used by Microsoft to load DLLs for testing and debugging purposes while developing Office applications. | | | | |
| Document | Description of attack technique 'T1137.002: Office Application Startup: Office Test':\nAdversaries may abuse the Microsoft Office "Office Test" Registry key to obtain persistence on a compromised system. An Office Test Registry location exists that allows a user to specify an arbitrary DLL that will be executed every time an Office application is started. This Registry key is thought to be used by Microsoft to load DLLs for testing and debugging purposes while developing Office applications. This Registry key is not created by default during an Office installation.\n\nThere exist user and global Registry keys for the Office Test feature, such as:\n\n * HKEY_CURRENT_USER\Software\Microsoft\Office test\Special\Perf\n* HKEY_LOCAL_MACHINE\Software\Microsoft \Office test\Special\Perf\n\nAdversaries may add this Registry key and specify a malicious DLL that will be executed whenever an Office application, such as Word or Excel, is started. | | | | |
| | **Base Emb, Base Gen** | **Base Emb, Tuned Gen** | **Tuned Emb, Base Gen** | **Tuned Emb, Tuned Gen** | **OpenAI Emb, GPT 4-o** |
| Generated answer | Adversaries may abuse the Microsoft Office 'Office Test' Registry key to obtain persistence on a compromised system. | The 'Office Test' Registry key is used by Microsoft to load DLLs for testing and debugging purposes while developing Office applications. | Adversaries may abuse the Microsoft Office 'Office Test' Registry key to obtain persistence on a compromised system. | The 'Office Test' Registry key is used by Microsoft to load DLLs for testing and debugging purposes while developing Office applications. | The purpose of the 'Office Test' Registry key is to allow a user to specify an arbitrary DLL that will be executed every time an Office application is started. |
| $d^*$ rank in context | 3 | 3 | 3 | 3 | 1 |
| Correctness score (hard) | 0.0 | 1.0 | 0.0 | 1.0 | 0.4 |
| Correctness reason | The actual output contradicts the expected output, stating it's used for persistence by adversaries instead of testing and debugging purposes by Microsoft. | The actual output perfectly matches the expected output, providing the same information without any omissions or contradictions. | The actual output contradicts the expected output, stating it's used for persistence by adversaries instead of testing and debugging purposes by Microsoft. | The actual output perfectly matches the expected output, providing the same information without any omissions or contradictions. | The actual output provides a partially correct purpose of the 'Office Test' Registry key but includes arbitrary user specification which is not present in the expected output, and omits the testing and debugging purposes mentioned in the expected output. |

Table 9 presents a case study in which the 'Base Emb, Tuned Gen' configuration is able to answer a question accurately even in the absence of $d^*$. That shows that fine-tuning the generation model on questions that cover a document, can allow the model to answer unseen questions about that document even when it is not presented in the LLM prompt context.

Table 9: Evaluation Case Study: When was MoleNet first observed in use?

| | Question | When was MoleNet first observed in use? |
|---|---|---|
| | True answer | MoleNet has been observed in use since at least 2019. |
| | Document | Description of attack software 'S0553: MoleNet': |
| | | MoleNet is a downloader tool with backdoor capabilities that has been observed in use since at least 2019. |

| | Base Emb, Base Gen | Base Emb, Tuned Gen | Tuned Emb, Base Gen | Tuned Emb, Tuned Gen | OpenAI Emb, GPT 4-o |
|---|---|---|---|---|---|
| **Generated answer** | I am sorry, I do not have the answer to the question. | MoleNet was first observed in use in 2019. | MoleNet is a downloader tool with backdoor capabilities that has been observed in use since 2019. | MoleNet has been observed in use since at least 2019. | I am sorry, I do not have the answer to the question. |
| $d^*$ **rank in context** | NaN | NaN | 0.0 | 0.0 | NaN |
| **Correctness score (hard)** | 0.0 | 0.8 | 0.9 | 1.0 | 0.0 |
| **Correctness reason** | The actual output does not provide any information about when MoleNet was first observed in use, whereas the expected output states it has been observed since at least 2019. | The actual output provides a specific year that matches the expected output, but it implies a specific start date, whereas the expected output leaves room for earlier usage with 'at least'. | The actual output provides the correct year MoleNet was first observed in use, but also includes irrelevant details about it being a downloader tool with backdoor capabilities. | The actual output matches the expected output exactly, with no omissions or contradictions. | The actual output does not provide any information about when MoleNet was first observed in use, whereas the expected output states it has been observed since at least 2019. |

## A.6 COMPARISON OF JUDGING CAPABILITIES OF LLAMA 3.1 405B AND GPT 4-O

In the spirit of promoting the use of SOTA open source LLMs throughout the solution pipeline, we used Meta's LLama 3.1 405B to judge the correctness of answers. Llama 3.1 405B performs at high speed (132 tokens/sec) and is offered for free use on SambaNova Cloud, making it an attractive alternative to proprietary LLMs with paid APIs.

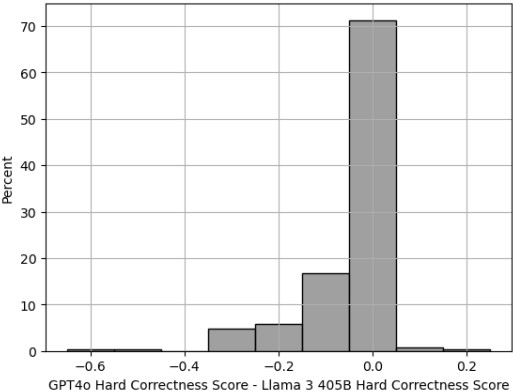

Figure 5: Distribution of the difference between hard correctness scores as judged by GPT 4-o and LLama 3.1 405B

The various pipelines in Table 3 were evaluated by LLama 3.1 405B and in, this appendix, we make a comparison between the scores generated by Llama 3.1 405B and GPT 4-o. A random sample of 400 answers, which were generated by the OpenAI pipeline (comprising Text Embedding 3 Large and GPT4-o), were evaluated in this comparison by Llama 3.1 405B and GPT 4-o on hard correctness. The same prompts were used for both judging models under the DeepEval GEval framework.

As shown in Fig. 5, on 71.25% of the sample, the two judging models gave exactly the same score. On 88.75% of the sample, the two models differed by 10 percentage points or less. On the minority of examples for which the two judging scores differed, GPT 4-o tended to be more strict, giving a lower score that explains the bars on the negative side of the difference.

The largest difference between the two judging models in terms of their scoring of the same generated answer is given in Table 10. In that example, we believe that GPT 4-o was too harsh in giving a score of zero and would have preferred the assessment given by LLama 3.1 405B.

Table 10: Evaluation Case Study: When was MoleNet first observed in use?

| | |
|---|---|
| **Question** | How many mitigation strategies can be used against attack technique 'T1216: System Script Proxy Execution'? Name them all. |
| **True answer** | The following 1 mitigation strategies mitigate attack technique 'T1216: System Script Proxy Execution': M1038: Execution Prevention |
| **Generated answer** | There are 2 mitigation strategies that can be used against attack technique 'T1216: System Script Proxy Execution': M1038: Execution Prevention, M1040: Behavior Prevention on Endpoint. |
| **Correctness score (GPT 4-o)** | 0.0 |
| **Correctness reason (GPT 4-o)** | The actual output contradicts the expected output by stating there are 2 mitigation strategies instead of 1 and includes an additional strategy, M1040: Behavior Prevention on Endpoint, which is not in the expected output. |
| **Correctness score (LLama 3.1 405B)** | 0.6 |
| **Correctness reason (LLama 3.1 405B)** | Actual output mentions two mitigation strategies, but expected output only mentions one, M1038: Execution Prevention. However, actual output does correctly identify M1038 as a mitigation strategy. |

In summary, both GPT 4-o and LLama 3.1 405B are suitable for judging tasks and they closely agree in $88.75\%$ cases. In the remaining cases, we found GPT 4-o to generate lower scores than we would have liked for answers that were also generated by GPT 4-o on AttackQA. Therefore, Llama 3.1 405B was not only a good choice for its better speed and cost, but also for fair evaluations of subjective answers.

