# OpenReview forum: "AttackQA: Development and Adoption of a Dataset for Assisting Cybersecurity Operations using Fine-tuned and Open-Source LLMs"
_ICLR.cc/2025/Conference — Submitted to ICLR 2025_

### Official Review · Reviewer_DQxm · 2024-11-01

**Soundness:** 3
**Presentation:** 2
**Contribution:** 2
**Rating:** 3
**Confidence:** 4

**Summary:**

This paper curated a dataset named AttackQA based on  MITRE ATT&CK knowledge base and fintune the LLM and embedding model to show that the performance of QA and retrieval can be improved by fine-tuning.

**Strengths:**

+ It takes some efforts to curate the dataset
+ The security dataset is needed

**Weaknesses:**

- The paper is a bit hard to follow for readers unfamiliar with the cybersecurity
I would say, for a machine learning conference, the paper should be able to be understood by general audience, instead of only readers who already have a strong background of cybersecurity. For example, in the related work section, the paper could expand on giving a detailed example of one entry of MITRE ATT&CK dataset since this work is based on that. The same for your dataset. You should provided detailed examples for your dataset, instead of leaving them in appendix. Although the paper mentions the  attack technique ’T1562.001' as example in the main body, it is hard for general readers to understand what is this for (I have some bg for cybersecurity but I also have no idea before searching for it). I would suggest replacing it with a more common CWE example such as SQL injection or integer overflow.

- The paper writing could be improved.
I think the paper writing could be enhanced if the authors put more efforts on that. For example, there are many double quotes like this ’T1562.001’. Please check how to properly use that in latex. Some figures like Fig4 is too small to see the texts on that. Moreover, the table alignment for some tables make it painful to read, especially Tab9.

- Irrelevant contribution about decoding speed.
This paper claims the high decoding speed by SambaNova Cloud platform, and mentions it a lot in paper. However, I don't think using a cloud platform can be deemed as contribution. When we refer to inference efficiency, we usually refer it to serving framework such as vllm, sglang, or attention mechanism such as flash attention 2, or decoding strategy such as lookup decoding. If you really want to show the overhead is small and this system can be real-time, you should at least apply these commonly used techniques since they are open-sourced.

- Unclear technique details
Due to the missing example, I found section 3.1-3.4 hard to understand. For example, what is your manual heuristics? why do you choose Llama-3-8B instead of gpt-4o or others? what is the instruction prompting? The missing of these details make readers hard to follow.

- Missing insights of model choice
The paper choose Llama-3 family for the whole work, while not even mentioning some sota open-sourced model such as Qwen-2.5-72B. There is no insights of such model prefernce. Also, in Section 3.5.2, the paper mentions that the Llama-3-70B(before finetuning) can outperform GPT-4o without any explanation nor detailed examples. It has no insights for the community.

- Missing training details
In section 4, a lot of training details for RAG for embedding model are missing.

- Choice of judgement model without justification
Again, this works chooses Llama-3 family (Llama-3-405B) for judgement. Typically, for the evaluation part, it needs justification for why such a method could suit. It won't be a problem if you pick gpt-4o since many LLM papers are doing that. It is commonly known as SOTA and able to judge. You CAN justify that gpt-4o is not accurate if you could provide detailed explanation and examples. Otherwise, it is not convincing to pick a 405B model for judgement as it is too large to run locally and unclear whether it is the SOTA method for judgement.

**Questions:**

See comments above

---

### Official Review · Reviewer_wewU · 2024-11-02

**Soundness:** 4
**Presentation:** 4
**Contribution:** 3
**Rating:** 6
**Confidence:** 4

**Summary:**

The paper proposes a dataset, AttackQA,
designed to assist analysts in Security Operations Centers (SOCs)
with timely and accurate answers to cybersecurity-related questions.
The dataset is created using the MITRE ATT&CK knowledge base
and features final 25,335 question-answer (Q&A) pairs fine-tuned to
improve retrieval-augmented generation (RAG) pipelines.
The proposed system is intended to streamline SOC workflows,
enabling analysts to quickly access high-quality,
contextualized information, thereby addressing key challenges in cybersecurity operations,
such as knowledge gaps and slow response times.

The reviewer acknowledges the relevance of the dataset and the potential impact of a fast,
reliable RAG pipeline that can be deployed using accessible,
open-source technology. The reviewer thinks this work is a valuable addition to the cybersecurity
field by providing a high-quality, open-source tool for cybersecurity-specific question answering.

However, the paper does not explain how this dataset could be useful for the cybersecurity research community. The reviewer recommends adding a discussion and/or subsection to discuss specific research applications and practical uses of AttackQA for cybersecurity researchers. This addition would help readers realize the dataset’s value and relevance for research in cybersecurity.

**Strengths:**

1. Domain-Specific Dataset

The reviewer notes the paper’s primary strength in creating a reasonably large,
cybersecurity-specific Q&A dataset that
leverages the MITRE ATT&CK knowledge base, a widely adapted resource in the cybersecurity community.
The paper has included detailed
descriptions of the dataset generation process,
including both human-generated and LLM-generated Q&A pairs,
and a quality control process.
The review believes this structured approach increases confidence in the dataset's
quality and applicability for SOC needs.

However, the reviewer wonders about the representativeness
of the Q&A pairs, particularly how well they reflect real SOC inquiries.
The authors might elaborate on how they determined which types of questions
would be most beneficial to SOC analysts.

2. Quality Control Mechanism

The reviewer appreciates the paper’s attention to quality control,
which involves fine-tuning LLMs (e.g., Llama 3 70B) to identify and
filter low-quality Q&A pairs.

The reviewer wonders if the paper could provide more
insight into how the quality control model was trained and
the rationale behind choosing G-Eval metric for
quality acceptance.


3. Model Deployment with High-Throughput Performance

The paper highlights that AttackQA models achieve high token generation speeds
using specialized hardware, such as the SambaNova Cloud,
which is particularly beneficial for SOC environments where latency is critical.

The reviewer wonders if the paper has considered testing the model
on consumer-grade hardware. Including performance benchmarks on less specialized
hardware could make the paper more broadly applicable to organizations with limited resources.

**Weaknesses:**

1. Limited Validation in Real-World SOC Settings

The reviewer observes that the dataset and RAG pipeline have not yet been validated
in an operational SOC environment,
limiting the understanding of AttackQA’s real-world effectiveness.

The reviewer suggests that a small-scale deployment or pilot study could greatly enhance
the paper’s credibility.
Even preliminary data on SOC analysts’ feedback or performance improvements in
simulated SOC scenarios would substantiate the system's value.



2. Dependence on Specialized Hardware

The reported performance relies on access to high-performance hardware,
such as the SambaNova Cloud, which may not be available to all SOCs.
This could limit the scalability and accessibility of the proposed solution.

To increase accessibility, the reviewer recommends that the authors benchmark AttackQA on
widely available hardware, such as NVIDIA consumer GPUs,
and report these results as part of the analysis.
This would demonstrate the adaptability of the pipeline to a range of environments.


3. Scope of RAG Framework

The reviewer notes that while the RAG framework is effective,
it may not fully address the complexity of real SOC inquiries that
require multi-hop reasoning or cross-document synthesis.
AttackQA, in its current form, may be limited in answering such complex questions accurately.

The reviewer recommends that the authors may consider extending the framework to
include multi-hop reasoning or explore alternative retrieval models that can
handle cross-document synthesis. This could make the model more adaptable to complex,
real-world SOC questions.

4. Error Analysis and Failure Case Discussion

The reviewer finds that the paper lacks an analysis of common failure cases,
which could help in understanding potential limitations of AttackQA and
provide directions for improvement.

The reviewer suggests conducting an error analysis to identify and
categorize common failure cases, such as instances where the model’s answers are incomplete,
incorrect, or hallucinated.
Outlining these challenges and discussing potential improvements would strengthen the paper.

5. Practicality and Utility in the Cybersecurity Research community

The paper does not discuss how AttackQA might be used by researchers,
leaving questions about its broader utility in the cybersecurity field.
Note that the reviewer is focusing on the cybersecurity research community, not its application in the industry.

- How could this dataset be useful for the cybersecurity research community?
- In what ways can researchers use this dataset for open science or further studies?
- What specific research applications and practical uses does AttackQA offer for cybersecurity researchers?
- How does the dataset add value and relevance to research in cybersecurity?

**Questions:**

1. How were the Q&A pairs chosen to ensure that they reflect the most relevant types of
questions SOC analysts would encounter?

2. Could the paper provide more details on the specific metrics, thresholds,
and methods used in the quality control process to filter Q&A pairs?

3. In what types of scenarios or question types does AttackQA outperform GPT-4o,
and could the paper elaborate on the specific strengths in these contexts?

4. Has the model’s performance been tested on consumer-grade hardware,
and if so, could these results be included?

5. Did the paper do any user studies or
tests with SOC analysts to gather real-world feedback
on the tool's performance and utility?

6. Could the paper discuss any potential performance
trade-offs or adjustments when deploying AttackQA on hardware with limited resources?

7. Did the paper consider testing AttackQA’s performance on
more complex, multi-step SOC questions?
 If so, what were the findings, and/or is this an area of future exploration?

9. Could the paper share preliminary findings on common
errors or failure cases, and what strategies might address these limitations?

10. How could this dataset be useful for the cybersecurity research community?

11. In what ways can researchers use this dataset for open science or further studies?

12.  What specific research applications and practical uses does AttackQA offer for cybersecurity researchers?

13. How does the dataset add value and relevance to research in cybersecurity?

---

### Official Review · Reviewer_1MEo · 2024-11-03

**Soundness:** 2
**Presentation:** 1
**Contribution:** 2
**Rating:** 3
**Confidence:** 4

**Summary:**

The paper introduces AttackQA, a cybersecurity Q&A dataset comprising 25,335 Q&A pairs with rationales, generated using the open-source LLM Llama-3 8B. A larger Llama-3 70B model was fine-tuned to improve dataset quality by filtering out low-quality pairs. The dataset was used to fine-tune open-source embeddings and LLMs, demonstrating better accuracy compared to OpenAI's GPT-4o. This work aims to support analysts in security operations centers by enhancing the efficiency and effectiveness of cybersecurity operations.

**Strengths:**

Main strengths:
+ Reduce both the cost and complexity of the system by utilizing open-source LLMs and embedding models.
+ Fine-tuning open-source LLM and embedded models achieves better performance than using proprietary models.

**Weaknesses:**

Main weaknesses:
-  Since AttackQA mainly focuses on textual knowledge, it cannot address the challenges of rapid response to cybersecurity incidents. In practice, Security Operations Centers (SoCs) also need to check  various tools' result to investigate the incidents.
- The experimental results may not be universally applicable, as AttackQA relies on a single dataset named MITRE ATT&CK® .
- The methodology for selecting techniques at each stage is not explained in sufficient detail.
- The experiments lack a comparison with existing solutions in this field.
- Partial grammatical errors and incomplete legends, such as the first paragraph in the conclusion section and  in the legend of Figure 2.

**Questions:**

1.  What is the performance of the existing Q&A systems for network security datasets, and  what advantages does AttackQA provide?

2. Given the abundance of cybersecurity datasets, why did the author choose MITRE ATT&CK® .  If AttackQA were trained on a different  dataset, would the performance vary significantly? How can AttackQA be scaled to other datasets?

3. How did the authors choose the LLM models through the entire process? What are the criteria and reasons for choosing LLM for different stages and tasks?

4. Could the authors elaborate on the specific techniques and strategies employed during the fine-tuning of the Llama 3 8B and Llama 3 70B models? What were the main challenges encountered, and how were they addressed?

5. In section 3.1, the paper claims each Q&A pair is derived from a single document because answers do not require information from multiple documents. Why? I question the validity of this assertion.

---

### Official Review · Reviewer_Rb91 · 2024-11-04

**Soundness:** 3
**Presentation:** 3
**Contribution:** 3
**Rating:** 5
**Confidence:** 3

**Summary:**

This paper presents a detailed process for dataset construction and model capability enhancement, centred around the topic of cybersecurity Q&A. The paper describes the experimental details to illustrate its reliability, and gives specific experimental results with case studies. When the fine-tuned models based on AttackQA are integrated into related RAG pipeline, it can effectively improve the model performance in cybersecurity domain.

**Strengths:**

- Build high-quality, reliable dataset AttackQA for cybersecurity community
- Plans for dataset and benchmark open-source
- In addition to the dataset generation, a significant amount of work has been done to illustrate feasible application scenarios for AttackQA, enriching the hierarchy of the whole paper

**Weaknesses:**

- The discussion of related datasets is missing in the Related Work section
- Lack of ethics review as a cybersecurity domain paper that might raise ethical concerns

**Questions:**

I appreciate the author's efforts throughout the paper, this work has a clear contribution and recursive structure, my main concern is the limitations of the contribution. Specifically, all of evaluations are based on AttackQA dataset, despite the author splits it to training set and the evaluation set, but they are also generated in the same source and have similar distribution. Then it is not surprising that the improvement in experimental results is achieved on the test set after fine-tune the model, the evaluation experiments can only show that the AttackQA fine-tuned model performs better on AttackQA after the model has been integrated into the RAG pipeline, but cannot show that the model has improved its overall performance in the cybersecurity domain. This paper is an excellent technical report, containing detailed parameters and specific results, but the overall contribution is not sufficient to support it as a research paper.

In addition, as a security-related paper that might raise ethical concerns, having a relevant discussion indicates that due diligence has been made to minimise potential harm has become a common practice, hope the authors can provide this information.


*minors*:
- Unify the terms. `The MITRE ATT&CK knowledge base` and `The MITRE ATT&CK@ knowledge base`
- Line173-175, the use of inverted commas is not standardised, it is recommended to present the dataset example by a separate text box rather than plain text

**Details Of Ethics Concerns:**

This paper used MITRE ATT&CK, which is a knowledge base of adversary tactics and techniques, to generate a dataset on cybersecurity, named AttackQA, but lacked a discussion addressing the harmfulness of the dataset.

---

### Meta-Review · Area_Chair_jQ7L · 2024-12-17

**Metareview:**

The paper presents AttackQA, a cybersecurity Q&A dataset with 25,335 question-answer pairs derived from the MITRE ATT&CK knowledge base and refined using open-source LLMs like LLaMA-3. Fine-tuning LLMs and embedding models with AttackQA improves performance in retrieval-augmented generation (RAG) pipelines, surpassing GPT-4o.

The paper has merits in building a domain-specific dataset for cybersecurity and improving model performance based on the dataset. But the paper still has weaknesses:
- The presentation is not good enough to illustrate the contributions.
- The model selection issue raised by multiple reviewers, that only LLaMA is used for comparison.
- The experiments are limited to the MITRE ATT&CK, which may not be generalizable.

The paper received mixed reviews with one borderline accept recommendation, one borderline reject recommendation, and two reject recommendations. After author-reviewer discussion and AC-reviewer discussion, the paper still has some weaknesses despite its contributions on building the cybersecurity dataset and LLM. Therefore, the AC considers that the paper falls short of the ICLR acceptance threshold and recommends rejection.

**Additional Comments On Reviewer Discussion:**

Reviewer Rb91 initially raised ethical concerns and concerns about limited contribution, and the authors addressed them. The reviewer had a follow-up concern regarding the benchmark in the paper, while it did not have a feedback. The AC agrees with the comment and thinks it needs to be addressed.

Reviewer 1MEo raised concerns about the performance and experimental setting. The authors tried to address them but failed to convince the reviewer.

Reviewer wewU initially raised concerns about Limited Validation in Real-World SOC Settings, Dependence on Specialized Hardware, Scope of RAG Framework, etc. The reviewer is generally in favor of this paper.

Reviewer DQxm also raised concerns about writing quality, irrelevant contribution, experimental designs. The authors also failed to satisfy the reviewers.

Based on the reviews and discussions, AC thinks that multiple points raised by the reviewers are not addressed and would recommend rejection.

---

### Decision · Program_Chairs · 2025-01-22

Reject